# Predictors of Abstinence from Smoking: A Retrospective Study of Male College Students Enrolled in a Smoking Cessation Service

**DOI:** 10.3390/ijerph16183363

**Published:** 2019-09-12

**Authors:** Yeji Lee, Kang-Sook Lee, Haena Kim

**Affiliations:** 1Department of Public Health, Graduate School, The Catholic University of Korea, 222 Banpo-daero, Seocho-gu, Seoul 06591, Korea; fina@catholic.ac.kr (Y.L.);; 2Department of Preventive Medicine, College of Medicine, The Catholic University of Korea, 222 Banpo-daero, Seocho-gu, Seoul 06591, Korea; 3Seoul Tobacco Control Center, 222 Banpo-daero, Seocho-gu, Seoul 06591, Korea

**Keywords:** smoking, college student, abstinence, smoking cessation intervention

## Abstract

There were high smoking rates among young male college students in Korea. We aimed to investigate the prevalence of and factors affecting abstinence from smoking following smoking cessation service attendance in this population. Data were collected between 1 August 2015 and 20 August 2018. Participants were administered more than nine face-to-face and telephone counseling sessions by trained tobacco cessation specialists for six months. Follow-up assessments were conducted at 4, 6, and 12 weeks, and 6 months after the quit date. A total of 3978 male college student smokers were enrolled; their mean age was 23.17 (±3.45) years. Almost one-third of the participants (64.9%) reported that they had attempted to quit during the past year. The number of cigarettes smoked per day, CO ppm, and Fagerström Test of Nicotine Dependence score at the baseline were negatively associated with abstinence, while motivational variables—importance, confidence, and readiness—were positively associated with abstinence. Our results suggest that provision of visiting smoking cessation services can be an active intervention platform for college student smokers who need professional assistance or support. Active and accessible support should be provided to such people. Visiting a smoking cessation service may result in increased long-term abstinence rates in such students.

## 1. Introduction

One of the major goals of tobacco control is reducing smoking among young adults, as quitting at an early age allows for the avoidance of the more harmful health consequences of smoking as well as tobacco-related mortality [1,2]. In Korea, the proportion of those receiving higher education, such as a college or university education, is 69.7%. Especially among young adults in their 20s, attendance in postsecondary institutions is high [3]. As of 2018, the number of university students in the city of Seoul was around 0.98 million, accounting for the greatest proportion (29%) of all university students in Korea (~3.4 million) [4]. The proportion of male young adult smokers (aged 19–29 years) was 37.3%—the greatest in the male smoker population [5]. Meanwhile, it was shown that there were high smoking rates among young male college students in Korea. According to the regional health assessment performed by the Korea Centers for Disease Control and Prevention, the proportion of male postsecondary student smokers was 30.0% in 2014 [6]. More specifically, the proportion of smokers was higher among students attending three or three-year programs (40.0%) than those attending four-year programs (27.0%) [6]. In addition, the starting age of lifetime smoking among people aged 19–29 years had steadily decreased to 17.8 years in male smokers [7]. Unsurprisingly, 65.2% of current young adult smokers (aged 19–29 years) have attempted smoking cessation, which is the highest across all age groups [5]. However, the majority of such young adults attempt quitting on their own, without professional assistance or support [8]. Active support from public health experts, and a more comprehensive approach aimed at motivating university students to quit smoking is required to prevent progression to moderate heavy smoking. In the provision of active intervention for smoking-related issues among young adults of university-going age, it is important to analyze the characteristics of their smoking habits and provide personalized smoking cessation programs, preferably at the campus. Although several universities have made their campuses smoke-free environments and provided smoking cessation clinics in cooperation with regional public health centers, there is a need for more systematic health promotional activities for the entire population of university students.

Since the establishment of the National Health Promotion Act in 1995—which includes various tobacco regulations, increases in the cost of cigarettes, expansion of smoke-free zones, regulation of tobacco advertisements, and provision of national-level smoking cessation programs (at public health centers and through telephone helplines)—the proportion of male adult smokers in Korea has been steadily decreasing, from 66.3% in 1998 to 38.1% in 2017 [5]. However, there is a blind zone associated with the fact that the focus groups of most of those smoking cessation programs comprised male adults. In 2016, the age group which showed the highest frequency of smoking cessation clinic attendance at public health centers was 30–59 years (59.8% of all attendees) [9]. There is a need for additional services that can contribute to smoking cessation and commitment in young adults. Consequently, the Ministry of Health and Welfare in Korea established regional tobacco control centers across the nation in 2015. These centers provide visiting smoking cessation services in campuses for college students who have difficulty utilizing the traditional smoking cessation program due to time constraints or social prejudice.

This study aimed to identify the smoking behaviors and factors related to the habit among college-going smokers enrolled in visiting smoking cessation services. In addition, we sought to identify the success rates of smoking cessation at 4, 6, and 12 weeks, and 6 months after the quit date, and analyze the factors affecting abstinence from smoking among male college students.

## 2. Materials and Methods

### 2.1. Study Population and Procedures

The secondary data used in this study were extracted from each participant’s electronic record in the smoking cessation service (nosmk.khealth.or.kr) from 1 August 2015 to 20 August 2018. Current smokers who enrolled in college or were on leave at the time of enrollment, including graduate students, who were willing to quit smoking completely and showed an intention to participate in the service for six months, were deemed eligible for participation. The primary data size was 4047; data from a total of 3974 people were analyzed, after the exclusion of 43 participants who had missing information and 30 without carbon monoxide measurements at the baseline.

In each of the participating university campuses in Seoul, the health services clinic promoted the smoking cessation service by putting up posters and sending students e-mails and SMSes. Participants enrolled in the service through awareness raised by smoke-free campaigns, the recommendation of other people, and the promotion of non-smoking buses distributed by the Ministry of Health and Welfare. Before enrollment in the service, students decided whether or not to participate in the service after being provided information on consent in accordance with the Personal Information Protection Act. Students who provided informed consent could enroll in the service and a registration card was created.

The service included more than nine face-to-face and telephone counseling sessions provided over a six-month period. Trained tobacco cessation specialists visited the campus and conducted face-to-face counseling. Participants could visit the Seoul Tobacco Control Center for face-to-face counseling under special circumstances. Depending on the outcome of each counseling session, follow-up face-to-face counseling sessions were provided, as measurements of exhaled carbon monoxide or indicated cotinine concentrations were required. Under inevitable circumstances, visiting counseling could be replaced by telephone counseling. Each counseling session included discussions on the development of problem-solving and coping skills, identification of withdrawal symptoms and provision of coping strategies, identification of strategies to cope with the urge to smoke and recognition of change after quitting, healthy behavior counseling, and relapse prevention. Nicotine replacement therapy (NRT) was provided to participants who had a Fagerström Test of Nicotine Dependence (FTND) score higher than 4, reported cigarettes per day CPD level ≥10 at the baseline, or would be willing to use NRT for themselves. Counselors provided with the principle of providing NRT by determining the use of nicotine supplements and nicotine content considering participants’ smoking quantity, degree of nicotine addiction, general health condition, and contradictions. Table 1 shows the manual for the counseling of each session.

This study was approved by the Institutional Review Board of the Catholic University of Korea (MC17EESI0090).

### 2.2. Measurement

#### 2.2.1. Registration Card and Assessment of Smoking History and Smoking Behavior

Baseline variables included age, exercise, drinking, and disease presence. In every enrolled participant, the level of exhaled carbon monoxide was measured at the baseline. Participants were asked about the number of cigarettes smoked per day, age at which the first cigarette was smoked, and number of quit attempts in the past year. In participants who answered that they had attempted to quit during the past year, the number of days in which they attempted to quit and the method of quitting were recorded. The method of quitting was divided into self-reliance and seeking professional help (i.e., visiting smoking cessation clinics at public health centers, using telephone helplines, taking help from a doctor or medical clinic, etc.). Participants were also asked about the reason that contributed the most to their desire to quit.

#### 2.2.2. Nicotine Dependence

Nicotine dependence was measured using the FTND score, the validity and reliability of which have been proven [10]. The scale ranges from 0 to 10. The higher the score, the greater the degree of nicotine addiction is. Nicotine dependency was classified into three groups according to the sum of the FTND scores: Light (0 ≤ FTND score ≤ 3), moderate (4 ≤ FTND score ≤ 6), and high (7 ≤ FTND score ≤ 10).

#### 2.2.3. Motivational Rulers

Participants rated three rulers associated with the motivation to quit smoking at the baseline [11,12]. Importance was indexed by, “How important is smoking cessation to you? (0 = Not important at all; 10 = Most important goal in my life).” Confidence was indexed by, “How confident are you that you could be successful in quitting smoking? (0 = Not confident at all; 10 = Most confident).” Readiness was indexed by, “How ready are you to quit? (0 = Not ready at all; 10 = Perfectly ready).” These rulers have a 0–10 scale.

#### 2.2.4. Follow-up Assessment Measurement

Abstinence data were collected at 4-week, 6-week, 12-week, and 6-month follow-ups (measured from the quit day). Follow-up assessments were preferably conducted via face-to-face counseling; in cases in which this was not possible, telephone counseling was used. Biochemical validation was only possible during face-to-face counseling when participants self-reported their degree of abstinence. Abstinence of biochemically validation was accessed when at least one of two methods of measuring exhaled carbon monoxide or urinary cotinine concentration was identified. For carbon monoxide, a level of 10 ppm or lower was indicative of the achievement of abstinence from smoking. In terms of the urine cotinine test, a negative result was recorded as the achievement of abstinence. The service was terminated if participants failed to quit while receiving the service or indicated an intention to stop participating in the service (e.g., moving to another region, etc.), or if contact was lost. Participants lost to follow-up or terminated were considered as continuing to smoke.

### 2.3. Statistical Analysis

Descriptive analysis was conducted to determine the baseline characteristics of all the participants. To compare the characteristics of the failed and successful quitters, categorical data were compared with a chi-square test, and continuous data with a t-test. Binary logistic regressions were performed for each predictor, as well as the rulers—importance, confidence, and readiness—with self-reported abstinence data at the 4-, 6-, 12-, and 24-week mark from the quit day as dependent variables, after controlling for age, exercise, drinking, and disease presence. These variables which were used to control within logistic regressions were considered as potential confounders [13,14,15]. All *p*-values were set at <0.05. All analyses were conducted with SPSS 25.0 (IBM, Armonk, NY, USA, 2017).

## 3. Results

### 3.1. Characteristics of Enrollees

Table 2 represents the baseline characteristics of the enrollees. There were 3974 participants in this study with 3643 six-month failed cases and 331 six-month successful quitters. The mean age of the participants was 23.17 (±3.45) years. Most participants (83.9%) responded that they consumed alcohol and only 2.7% stated that they had a disease. Almost one-third of the participants (64.9%) reported that they had attempted to quit smoking during the past year. Low level of nicotine dependence was observed in more than half of the participants (60.8%), and those who were successful in quitting smoking for six months were likelier to have a lower FTND score (2.12 ± 2.10 vs. 3.30 ± 2.12, *p* < 0.001) than those who were not. At the baseline, the mean concentration of carbon monoxide was 10.95 (±7.41) ppm, the average number of cigarettes smoked per day was 11.86 (±6.30), and the age at which the first cigarette was smoked was 18.34 (±2.72) years. Participants who were successful in smoking cessation for six months were likelier to have undergone a higher number of counseling sessions (6.51 ± 2.45 vs. 1.70 ± 1.11, *p* < 0.001) than those who were not.

### 3.2. Smoking Abstinence Prevalence

Table 3 shows the smoking abstinence prevalence at the 4-, 6-, and 12-week, and 6-month marks, as observed through self-reports and biochemical validation. The self-reported abstinence prevalence includes biochemically validated abstinence prevalence. The self-reported abstinence prevalence rates at the 4-, 6-, and 12-week, and 6-month marks were 17.5%, 14.9%, 10.8%, and 8.3%, respectively. The biochemically validated abstinence prevalence rates at the 4-, 6-, and 12-week, and 6-month marks were 8.7%, 7.3%, 3.3%, and 4.4%, respectively.

### 3.3. Predictors of Abstinence

Table 4 represents the predictors of abstinence at 4, 6, and 12 weeks, and 6 months. The AORs for past year quit attempts were 1.34, 1.29, 1.32, and 1.40 for participants who were abstinence at 4 weeks, 6 weeks, 12 weeks, and 6 months, respectively. Specially, past year quit attempts were proven to be the best predictor of quitting (OR: 1.40, 95% CI: 1.09–1.79). With each unit increase in the number of cigarettes smoked per day at the baseline, the AORs for abstinence maintenance at 4 weeks, 6 weeks, 12 weeks, and 6 months were 0.93, 0.94, 0.93, and 0.92, respectively. In addition, with each one-unit increase in the CO ppm at the baseline, the ORs of abstinence maintenance at 4 weeks, 6 weeks, 12 weeks, and 6 months were 0.90, 0.90, 0.88, and 0.87, respectively. For each increase in the FTND score, the ORs of abstinence maintenance at 4 weeks, 6 weeks, 12 weeks, and 6 months were 0.86, 0.85, 0.83, and 0.81, respectively.

### 3.4. Motivational Rulers Predicting Abstinence

Table 5 shows the odds ratios for motivational rulers predicting abstinence maintenance at 4, 6, and 12 weeks, and 6 months. With each unit increase in the reported level of importance given to quitting at the baseline, the AORs for abstinence maintenance at 4 weeks, 6 weeks, 12 weeks, and 6 months were 1.10, 1.08, 1.09, and 1.11, respectively. For each one-unit increase in the reported confidence level in quitting at the baseline, the AORs for abstinence maintenance at 4 weeks, 6 weeks, 12 weeks, and 6 months were 1.23, 1.24, 1.26, and 1.29, respectively. In addition, for each additional score in terms of the readiness to stop smoking, the AORs for abstinence maintenance at 4 weeks, 6 weeks, 12 weeks, and 6 months were 1.24, 1.25, 1.28, and 1.32, respectively.

## 4. Discussion

Smoking cessation services offer continuity of care, long-term counseling services, and support to college-going smokers who want to quit. Our study proved that the report of a greater number of cigarettes smoked per day was associated with a lower abstinence rate at the follow-up. Quit and Win contests among college students, which were based on financial incentives, have found that the number of cigarettes smoked per day predicted abstinence maintenance at six months (odds ratio = 0.92) [16]. A recent study by Ni et al. reported that, while the categorization of smokers into groups based on their cigarette consumption level was not independently associated with the smoking cessation rate, it affected their confidence in quitting—an important factor that predicts abstinence [17]. Due to older age being associated with a greater number of cigarettes smoked among young adults [14,18,19,20], smoking cessation interventions must be provided at a young age.

Consistent with previous studies, which showed that a stronger nicotine dependency was negatively associated with a failure to stop smoking [21,22], our study revealed that a stronger nicotine dependency at the baseline led to lower abstinence rates. Severe nicotine dependency can make it difficult for smokers to quit smoking due to withdrawal symptoms that are related to relapse [23]. In this study, 60.8% of the participants showed slight nicotine dependency; therefore, it is crucial to help college-going smokers who have not yet developed a strong nicotine dependency to quit smoking before the degree of dependence worsens and provide them with the motivation to quit. In addition, college-going smokers with a strong nicotine dependency must adapt effective and evidence-based programs for relief from withdrawal symptoms, such as NRT and motivational intervention [24,25].

The longer the smoking period, the greater the amount of cigarette smoking is, and the higher the amount of cigarette smoking, the stronger the nicotine dependence. Thus, early smoking interventions must be provided to college-going smokers before their period of smoking and amount of cigarettes smoked increase, and progression to a more severe level of dependence occurs.

This study found that higher scores in the three motivational rulers (importance, readiness, and confidence) were associated with greater odds of abstinence at the follow-up. Motivational rulers are widely used in clinical settings [26], and Chung et al. revealed that their measurement at the baseline could predict the average number of cigarettes smoked per day at the 12-month follow-up [27]. Boudreaux et al. found that motivational rulers significantly predict smoking-related behavioral change, but none of the motivational variables could differentiate between those who tried to quit but relapsed and those who successfully quit [12]. This study provides support for the assessment of importance, readiness, and confidence in smoking cessation counseling for college students as predictors of abstinence. In particular, all the motivational variables showed the strongest associations in the prediction of abstinence at six months. Especially, readiness was proven to be the best predictor of quitting in the motivational variables (OR: 1.32, 95% CI: 1.25–1.34). It suggests that intervention in this program might help enrollees who have the importance, readiness, or confidence in quitting that the counselors help them with problem solving, coping, identification of withdrawal symptoms and ability to cope with the urge to smoke, and strengthening their motivation to quitting.

It is noteworthy that 64.9% of the participants attempted to quit smoking in the past year. This result is similar to that observed in recently obtained nationally representative Korean data [5], which showed that young adults (aged 19–29 years) have the highest rate of smoking attempts (65.2%) across all age groups. Solberg et al. found that young adults were likelier to have attempted to quit smoking in the last year than older adults [5]. Messer et al. revealed that the proportions of those who made serious attempts at quitting in one year, quit for at least one day. Those who quit for at least six months were the highest among adults aged 18–24 years [28]. Our study demonstrated that those who attempted to quit during the last year eventually enrolled in the service after an average of 63.57 (±87.62) days of abstinence.

However, only 8.1% of the attempters received professional help, with most of them making attempts without medication or counseling assistance. This is consistent with the results of a previous study conducted in Sweden in which college-going smokers reported they were less likely to use professional resources such as national telephone quit-lines [29]. Young adult smokers are less likely to seek aid from health professionals in smoking cessation than older adult smokers [5,27]. Zhu et al. found that the rates of use of assistance for smoking cessation increased with age [22]. In addition, contrary to the assumption that college students wanted to quit due to the increase in cigarette prices since 2015, only 12.9% of our participants reported that they wanted to quit for financial reasons. More than half of the participants (55.2%) in the service wanted to quit because they were concerned about their health; they wanted to prevent current disease progression and future disease occurrence. However, they were unaware of how to access professional aid. Their report of seeking professional help for quitting suggests that the provision of accessible, appropriate, and active support may decrease the smoking rate among college students.

This study reported that those who had attempted to quit smoking in the past year were likelier to show abstinence at 4, 6, and 12 weeks, and 6 months. This finding has implications for college health advocates, adding to a growing body of evidence which suggests that visiting a smoking cessation service may enhance engagement in quitting trials. Future cessation attempts using visiting smoking cessation services may enhance long-term abstinence through the reception of professional help such as medication prescription or therapeutic counseling. The increase in the rate of smoking cessation attempts may eventually lead to lower quit rates among students.

This study has some limitations. First, we did not recruit a control group to compare the effectiveness of visiting smoking cessation programs for college-going smokers. Second, the current study’s participants included university students who resided in Seoul—the capital of Korea; therefore, the findings may not generalize the total population of Korean university students. Third, the follow-up rate of this study was low, as the participants’ data were not available when termination occurred during the service. The fact that participants who were lost to follow-up or terminated were considered as continuing to smoke can cause bias. Fourth, as this study utilized secondary data, there were a limited number of variables that could be analyzed. Thus, this study was not able to accurately control for unobserved potential confounders. Fifth, logistic regression was performed using self-reported abstinence data, which has a lack of sufficient power compared to biochemically validated abstinence data. Sixth, only male college students were analyzed in this study. Further studies are needed to analyze both male and female college students. Nevertheless, this is the first retrospective study, including a large number of college-going smokers, to analyze six-month follow-up data obtained from a program conducted for three years, which was closely monitored by trained tobacco cessation specialists. Moreover, this study compared the characteristics of the failed and successful quitters, and it was proven that participants who were successful in smoking cessation for six months were likelier to have undergone a higher number of counseling sessions than those who were not. This suggests that counseling sessions in this study might help smokers quit.

Young adult smokers show the highest smoking rates, and college-going smokers comprise a high proportion of this population; however, no representative studies for this population in Korea have been yet conducted. Since 2007, the Ministry of Health and Welfare has been providing a non-smoking supporter system (http://www.nosmokeguide.or.kr/), which is a voluntary college student group that strives to create smoke-free campuses across universities nationwide. However, it is insufficient to merely provide financial support for smoking cessation counseling or education to individuals or through student clubs. It was found that health managers at most college health centers in the United States perceived smoking as a serious problem for college students within the campus, but nearly half of the colleges did not offer any smoking cessation programs for their students [30]. Therefore, much effort should be made to foster environmental support in the campus to increase engagement with cessation services and improve cessation rates. To raise awareness for the smoking-related issues among young adult population, the provision of smoke-free campaigns is needed. Moreover, proper and effective smoking cessation programs for college students within the campus should be implemented. The provision of visiting smoking cessation services, access to trained tobacco cessation specialists, and face-to-face counseling is needed to address the increasing demand for preventive services.

## 5. Conclusions

Smoking rates among male college students are high, but there is a low engagement with smoking cessation services among college-going smokers. The majority of college student smokers want to quit smoking, but there is a low engagement with smoking cessation services among college-going smokers. Provision of visiting smoking cessation services can be an active intervention platform for college student smokers who need professional assistance or support. The smoking prevalence among college students could decrease if college-going smokers are provided more accessible and professional support.

## Figures and Tables

**Table 1 ijerph-16-03363-t001:** Manual for counseling.

Session	Schedule	Topics
1	Enrollment (Day 1)	- Providing information on consent - Writing registration card - Assessment of smoking history and smoking behavior - Providing Fagerström Test of Nicotine Dependence (FTND) - Decision of the quit day - Counseling for strengthening motivation to quit smoking - Development of problem solving and coping skills - Providing Nicotine Replacement Therapy (NRT)
2	Abstinence at 1-week (Day 7)	- Check and support maintenance of smoking cessation - Identification of withdrawal symptoms and ability to cope with the urge to smoke - Evaluating the suitability of using NRT
3	Abstinence at 2-week (Day 14)	- Check and support maintenance of smoking cessation - Counseling for chronic diseases - Monitoring adverse events of using NRT
4	Abstinence at 4-week (Day 28)	- Validation of abstinence at 4-week (measuring exhaled carbon monoxide/urine cotinine test) - Evaluating the effectiveness of using NRT
5	Abstinence at 6-week (Day 42)	- Validation of abstinence at 6-week (measuring exhaled carbon monoxide/urine cotinine test) - Counseling for relapse prevention
6	Abstinence at 8-week (Day 56)	- Check and support maintenance of smoking cessation - Finding strategies against the urge to smoke - Overcoming enemies inside
7	Abstinence at 12-week (Day 84)	- Validation of abstinence at 12-week (measuring exhaled carbon monoxide/urine cotinine test) - Recognition of changes after quitting - Counseling for healthy behaviors
8	Abstinence at 16-week (Day 112)	- Check and support maintenance of smoking cessation - Counseling for relapse prevention
9	Abstinence at 6-month (Day 168)	- Validation of abstinence at 6-month (measuring exhaled carbon monoxide/urine cotinine test) - Providing a souvenir for a successful quitting

**Table 2 ijerph-16-03363-t002:** Baseline characteristics of college-going smokers in the Visiting Smoking Cessation Services Program in Korea, 2015 to 2017 (*N* = 3974).

Variable	All Enrollees (*n* = 3974)	6-Month Failed Cases (*n* = 3643)	6-Month Successful Quitters (*n* = 331)	t-test or χ^2^	*p*-Value
	Mean ± SD or N (%)				
Age (y)	23.17 ± 3.45	23.15 ± 3.49	23.41 ± 2.99	−1.306	0.191
Exercise					
No	1561 (39.3)	1451 (39.8)	110 (33.2)	5.537	0.019
Yes	2413 (60.7)	2192 (60.2)	221 (66.8)		
Drinking					
No	639 (16.1)	578 (15.9)	61 (18.4)	1.477	0.244
Yes	3335 (83.9)	3065 (84.1)	270 (81.6)		
Disease presence					
No	3867 (97.3)	3545 (97.3)	322 (97.3)	0.001	0.975
Yes	107 (2.7)	98 (2.7)	9 (2.7)		
Past year quit attempt					
No	1396 (35.1)	1303 (35.8)	93 (28.1)	7.834	0.005
Yes	2578 (64.9)	2340 (64.2)	238 (71.9)		
Average days of smoking cessation *	63.57 ± 87.62	59.85 ± 84.18	100.10 ± 109.79	−5.494	<0.001
Method of smoking cessation *					
Seeking professional help	219 (8.5)	2147 (91.8)	212 (89.1)	1.991	0.158
Without professional help	2359 (91.5)	193 (8.2)	26 (10.9)		
FTND score	2.93 ± 2.18	3.30 ± 2.12	2.12 ± 2.10	7.131	<0.001
Nicotine dependency					
Low	2416 (60.8)	2170 (59.6)	246 (74.3)	27.901	<0.001
Moderate	1306 (32.9)	1233 (33.8)	73 (22.1)		
Severe	252 (6.3)	240 (6.6)	12 (3.6)		
CO ppm	10.95 ± 7.41	11.36 ± 7.41	6.42 ± 5.72	14.657	<0.001
Cigarettes smoked per day	11.86 ± 6.30	12.09 ± 6.21	9.41 ± 6.69	7.464	<0.001
Age at which first cigarette was smoked	18.34 ± 2.72	18.31 ± 2.71	18.70 ± 2.79	−2.504	0.012
Reason for quitting					
Recommendations from others	736 (18.5)	675 (18.5)	61 (18.4)	4.064	0.255
Concerned about health	2192 (55.2)	2010 (55.2)	182 (55.0)		
For financial reason	511 (12.9)	477 (13.1)	34 (10.3)		
For other reasons (None of these)	535 (13.5)	481 (13.2)	54 (16.3)		
Readiness Ruler					
Importance	6.96 ± 2.12	6.92 ± 2.13	7.39 ± 2.10	−3.876	<0.001
Confidence	5.66 ± 2.42	5.54 ± 2.39	6.95 ± 2.30	−10.262	<0.001
Readiness	5.86 ± 2.33	5.74 ± 2.31	7.17 ± 2.09	−11.721	<0.001
Number of counseling sessions					
Face-to-face	1.76 ± 1.50	1.47 ± 0.92	4.93 ± 2.63	−23.841	<0.001
Telephone	0.35 ± 0.70	0.24 ± 0.52	1.58 ± 1.12	−21.722	<0.001
Total	2.10 ± 1.84	1.70 ± 1.11	6.51 ± 2.45	−35.335	<0.001
NRT					
No	2778 (69.9)	2533 (69.5)	245 (74.0)	2.904	0.088
Yes	1196 (30.1)	1110 (30.5)	86 (26.0)		

* *n* = 2578 (participants who answered yes to ‘Past Year Quit Attempt’). SD, standard deviation; NRT, nicotine replacement therapy; FTND, Fagerström Test of Nicotine Dependence.

**Table 3 ijerph-16-03363-t003:** Smoking abstinence prevalence rates at 4 weeks, 6 weeks, 12 weeks, and 6 months, as assessed by self-report and biochemical validation.

	4 Weeks	6 Weeks	12 Weeks	6 Months
Self-report	17.5%	14.9%	10.8%	8.3%
Biochemical validation	8.7%	7.3%	3.3%	4.4%

**Table 4 ijerph-16-03363-t004:** Adjusted odds ratios and 95% Confidence Intervals from the logistic regression models for the factors affecting smoking abstinence.

Factor	Odds Ratio (95% Confidence Interval)
*4 weeks*	
Past Year Quit Attempts	1.34 (1.13, 1.61)
Age at which first cigarette was smoked: 1-y increase	1.02 (0.99, 1.05)
Cigarettes smoked per day: One-unit increase	0.93 (0.92, 0.95)
CO ppm: One-ppm increase	0.90 (0.89, 0.92)
FTND score: One-unit increase	0.86 (0.82, 0.89)
*6 weeks*	
Past Year Quit Attempts	1.29 (1.06, 1.55)
Age at which first cigarette was smoked: 1-y increase	1.02 (0.98, 1.05)
Cigarettes smoked per day: One-unit increase	0.94 (0.92, 0.95)
CO ppm: One-ppm increase	0.90 (0.88, 0.91)
FTND score: One-unit increase	0.85 (0.81, 0.88)
*12 weeks*	
Past Year Quit Attempts	1.32 (1.06, 1.64)
Age at which first cigarette was smoked: 1-y increase	1.03 (0.99, 1.07)
Cigarettes smoked per day: One-unit increase	0.93 (0.91, 0.95)
CO ppm: One-ppm increase	0.88 (0.87, 0.90)
FTND score: One-unit increase	0.83 (0.79, 0.87)
*6 months*	
Past Year Quit Attempts	1.40 (1.09, 1.79)
Age at which first cigarette was smoked: 1-y increase	1.05 (1.01, 1.10)
Cigarettes smoked per day: One-unit increase	0.92 (0.90, 0.94)
CO ppm: One-ppm increase	0.87 (0.85, 0.89)
FTND score: One-unit increase	0.81 (0.76, 0.86)

Adjusted by age, exercise, drinking, and disease presence. FTND, Fagerström Test of Nicotine Dependence.

**Table 5 ijerph-16-03363-t005:** Adjusted odds ratios and 95% Confidence Intervals from the logistic regression models for the motivational rulers affecting smoking abstinence.

Motivational Rulers	Odds Ratio (95% Confidence Interval)	*p*-Value
*Abstinence at 4 weeks*		
Importance	1.10 (1.05, 1.14)	<0.001
Confidence	1.23 (1.19, 1.28)	<0.001
Readiness	1.24 (1.20, 1.29)	<0.001
*Abstinence at 6 weeks*		
Importance	1.08 (1.04, 1.13)	<0.001
Confidence	1.24 (1.20, 1.29)	<0.001
Readiness	1.25 (1.20, 1.30)	<0.001
*Abstinence at 12 weeks*		
Importance	1.09 (1.04, 1.15)	<0.001
Confidence	1.26 (1.20, 1.32)	<0.001
Readiness	1.28 (1.22, 1.34)	<0.001
*Abstinence at 6 months*		
Importance	1.11 (1.05, 1.18)	<0.001
Confidence	1.29 (1.23, 1.36)	<0.001
Readiness	1.32 (1.25, 1.39)	<0.001

Adjusted for age, exercise, drinking, and disease presence.

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
