# Peer review of "Predictors of Abstinence from Smoking: A Retrospective Study of Male College Students Enrolled in a Smoking Cessation Service"

_ijerph, 2019, doi:10.3390/ijerph16183363_

Round 1
Reviewer 1 Report
The research part was planned properly and performed correctly. The description of methodology of the research is elaborate. Authors presented results of the study in an organized manner which was enhanced by precise tabular illustration. I value highly their ability of clear summary and interpretation of the outcomes in consideration of topical literature.
The study appears to be interesting not only from the scientific point of view, but the public health aspect should be emphasized.
To conclude, I can state that the assumed aim of the study has been accomplished. The article is written in an accurate, transparent style, using correct adequate terminology.
Author Response
Response to Reviewer 1 Comments
Point 1: The research part was planned properly and performed correctly. The description of methodology of the research is elaborate. Authors presented results of the study in an organized manner which was enhanced by precise tabular illustration. I value highly their ability of clear summary and interpretation of the outcomes in consideration of topical literature.
The study appears to be interesting not only from the scientific point of view, but the public health aspect should be emphasized.
To conclude, I can state that the assumed aim of the study has been accomplished. The article is written in an accurate, transparent style, using correct adequate terminology.
Response 1: The authors thank the reviewer for valuable comments. Thank you for your insights.

Reviewer 2 Report
This is an excellent study.
Introduction: excellent compact summary of the literature relating to S. Korea.
Intervention: It would be be helpful if the authors could describe in detail the manual or protocol for the counsellors to help with problem solving, coping, identification of withdrawal symptoms and ability to cope with the urge to smoke, and in enough detail future researchers can apply them.
Design: It is a pity there is no control group. This needs much more emphasis in the report.
Results: The authors need to emphasise the key results. The best predictor of quitting is prior quit attempts OR (1.40 (1.09 1.79 ) with a wide 95%CI so the predictor could add 9% to 79% to current quitting success. Also confidence OR 1.29 (1.23, 1.36) and readiness 1.32 (1.25 1.34) and the authors need to explain how their interventions could have effected these results.
The authors have reported the study in an intention to quit format (3643 failed, 331 quit) and they need to emphasise this.
I have said major revision. No change needed in data but change needed in write up.
Author Response
Response to Reviewer 2 Comments
This is an excellent study.
Point 1: Introduction: excellent compact summary of the literature relating to S. Korea.
Response 1: We thank all comments of the reviewer, and try to comply with suggestions. Replies were performed point by point. The included changes are highlighted in red color in the updated version of the manuscript.
Point 2: Intervention: It would be be helpful if the authors could describe in detail the manual or protocol for the counsellors to help with problem solving, coping, identification of withdrawal symptoms and ability to cope with the urge to smoke, and in enough detail future researchers can apply them.
Response 2: The authors have added Table 1 in the Materials and Methods section. We have described in detail the manual for the counselling of each session. We hope you find these explanations satisfactory.
Point 3: Design: It is a pity there is no control group. This needs much more emphasis in the report.
Response 3: We apologize for the confusion and thank you for highlighting this problem. In response to this comment, we have added more detailed description in the discussion section.
Point 4: Results: The authors need to emphasise the key results. The best predictor of quitting is prior quit attempts OR (1.40 (1.09 1.79 ) with a wide 95%CI so the predictor could add 9% to 79% to current quitting success. Also confidence OR 1.29 (1.23, 1.36) and readiness 1.32 (1.25 1.34) and the authors need to explain how their interventions could have effected these results.
Response 4: As the reviewer mentioned, we have emphasized several key results of this study with more detailed explanation in the result and discussion section.
Point 5: The authors have reported the study in an intention to quit format (3643 failed, 331 quit) and they need to emphasise this.
Response 5: We thank the reviewer for highlighting this important point. We provided more detail in the result section.
Point 6: I have said major revision. No change needed in data but change needed in write up.
Response 6: Thank you again for your constructive comments. We hope that this revised manuscript meets your expectations.

Reviewer 3 Report
This is an interesting paper, reporting the predictors of abstinence from smoking in male college students enrolled in a smoking cessation service. I think the paper would benefit from addressing the below comments:
Abstract
I’m not sure that the first sentence of the abstract, ‘there is a lack of efficient smoking cessation programs for college-going smokers in Korea’ is relevant to the study aims and existing literature in this area. I recommend changing this to summarise more relevant background literature (for example, high smoking rates among young male college students) Line 22 would be clearer if it read ‘positively associated with abstinence’ rather than ‘with the same’ Lines 22-23, ‘our results suggest that college-going smokers are unaware of how to utilize smoking cessation aid’. I do not think that you can make this conclusion/statement based on the aims, results and findings of your study. It may be more appropriate to comment on low engagement with smoking cessation services among your study population.
Introduction
Line 29, would be clearer reworded as ‘reducing smoking among young adults’ There is no justification either in the introduction, or elsewhere in the paper, why the study focuses only on males. This is a significant weakness that needs justifying, and discussing as a limitation of the paper in the discussion.Statistical analysis/results
What was the justification for choosing age, exercise, drinking and disease presence when deciding which variables to control for within logistic regression models?Discussion
The first paragraph of the discussion focuses on the finding that participants who initiated smoking at an older age had greater odds of abstinence. However, looking at table 3, this was only significant at 12 month follow-up, and only very marginally. I would recommend reconsidering discussing this as a major finding in the first paragraph of the discussion, and possibly recommend even discussing it at length within the discussion at all, given it is only a relatively small finding at one follow-up time point. Page 8, line 16-18 – the first sentence of this paragraph is not needed, this is an obvious finding (people who have been abstinent for longer are more likely to have more days abstinent from smoking), and is not relevant to the rest of the paragraph. Limitations – please discuss the study’s focus on males, and exclusion of females. Generally the discussion needs to focus more on how the findings can inform future cessation services/research to increase engagement with cessation services and improve cessation rates. The conclusion needs reconsidering, as I don’t think at the moment it reflects the study’s findings, or makes recommendations as to how to move forward given the new information this study provides. For example, is there any way that your findings can inform new ways to engage this population with cessation services?Author Response
Response to Reviewer 3 Comments
Point 1: This is an interesting paper, reporting the predictors of abstinence from smoking in male college students enrolled in a smoking cessation service. I think the paper would benefit from addressing the below comments:
Response 1: Thank you very much for your careful review and constructive comments. Following your recommendations, the authors have modified the manuscript. All replies were commented point by point. The included changes are highlighted in red color in the updated version of the manuscript.
Abstract
Point 2: I’m not sure that the first sentence of the abstract, ‘there is a lack of efficient smoking cessation programs for college-going smokers in Korea’ is relevant to the study aims and existing literature in this area. I recommend changing this to summarise more relevant background literature (for example, high smoking rates among young male college students)
Response 2: The authors thank the reviewer for their comments, and we revised the sentence to reflect relevant background literature.
Point 3: Line 22 would be clearer if it read ‘positively associated with abstinence’ rather than ‘with the same’
Response 3: We agree with the reviewer’s suggestion, and have revised the sentence correctly.
Point 4: Lines 22-23, ‘our results suggest that college-going smokers are unaware of how to utilize smoking cessation aid’. I do not think that you can make this conclusion/statement based on the aims, results and findings of your study. It may be more appropriate to comment on low engagement with smoking cessation services among your study population.
Response 4: We agree with the reviewer’s advice and have, therefore, revised the abstract to better reflect the aim of this research.
Introduction
Point 5: Line 29, would be clearer reworded as ‘reducing smoking among young adults’
Response 5: We agree with the reviewer’s suggestion and have revised the sentence accordingly.
Point 6: There is no justification either in the introduction, or elsewhere in the paper, why the study focuses only on males. This is a significant weakness that needs justifying, and discussing as a limitation of the paper in the discussion.
Response 6: We apologize for the confusion and thank you for highlighting this problem. In response to this comment, we have added more detailed description in the introduction and limitations section.
Statistical analysis/results
Point 7: What was the justification for choosing age, exercise, drinking and disease presence when deciding which variables to control for within logistic regression models?
Response 7: We apologize for the confusion and thank you for highlighting this problem. These variables were taken into consideration as potential confounders. We have added the commentary in statistical analysis sections.
Discussion
Point 8: The first paragraph of the discussion focuses on the finding that participants who initiated smoking at an older age had greater odds of abstinence. However, looking at table 3, this was only significant at 12 month follow-up, and only very marginally. I would recommend reconsidering discussing this as a major finding in the first paragraph of the discussion, and possibly recommend even discussing it at length within the discussion at all, given it is only a relatively small finding at one follow-up time point.
Response 8: We apologize for the confusion and thank you for highlighting this problem. In response to this comment, we deleted the following sentences:
“This study revealed that participants who initiated smoking at an older age had greater odds of abstinence. Early smoking initiation is a significant predictor of increased nicotine dependency [13], progression to heavy smoking [14], continuation of smoking in later life [13,15], and having difficulties in quitting smoking [13].”
Point 9: Page 8, line 16-18 – the first sentence of this paragraph is not needed, this is an obvious finding (people who have been abstinent for longer are more likely to have more days abstinent from smoking), and is not relevant to the rest of the paragraph.
Response 9: We apologize for the confusion and thank you for highlighting this problem. In response to this comment, we deleted the following sentences:
“Among participants who attempted to quit during the last year, the average number of days of abstinence in the last year was significantly longer in those who had abstained for 6 months than those who had not (100.10±109.79 vs. 59.85±84.18, p<.001).”
Point 10: Limitations – please discuss the study’s focus on males, and exclusion of females.
Response 10: We apologize for the confusion and thank you for highlighting this problem. In response to this comment, we have added more detailed description in the limitations section.
Point 11: Generally the discussion needs to focus more on how the findings can inform future cessation services/research to increase engagement with cessation services and improve cessation rates.
Response 11: In response to this comment, we have added more detailed description in the discussion section to focus the findings of this study.
Point 12: The conclusion needs reconsidering, as I don’t think at the moment it reflects the study’s findings, or makes recommendations as to how to move forward given the new information this study provides. For example, is there any way that your findings can inform new ways to engage this population with cessation services?
Response 12: We agree with the reviewer’s advice and have revised the entire conclusions to provide a better description.

Round 2
Reviewer 2 Report
The authors have responded to all the suggestions of the reviewers.